# Precision Medicine in Non-Communicable Diseases

**DOI:** 10.3390/ht9010003

**Published:** 2020-02-07

**Authors:** Giuseppe Novelli, Michela Biancolella, Andrea Latini, Aldo Spallone, Paola Borgiani, Marisa Papaluca

**Affiliations:** 1Department of Biomedicine & Prevention, Genetics Unit, University of Rome “Tor Vergata”, 00133 Rome, Italy; latini.andrea@hotmail.com (A.L.); borgiani@med.uniroma2.it (P.B.); 2IRCCS Neuromed, 86077 Pozzilli (IS), Italy; 3Department of Pharmacology, School of Medicine, University of Nevada, Reno, NV 89557, USA; 4Department of Biology, University of Rome “Tor Vergata”, 00133 Rome, Italy; michelabiancolella@gmail.com; 5Department of Neurology and Neurosurgery, Peoples’ Friendship University of Russia (RUDN University), Moscow 117198, Russia; aldospallone@hotmail.com; 6Imperial College, Faculty of Medicine, School of Public Health, SW7 2AZ London, UK; mpapaluca@me.com

**Keywords:** precision medicine, genomics, epigenetics, pharmacogenetics, genomic biomarkers

## Abstract

The increase in life expectancy during the 20th century ranks as one of society’s greatest achievements, with massive growth in the numbers and proportion of the elderly, virtually occurring in every country of the world. The burden of chronic diseases is one of the main consequences of this phenomenon, severely hampering the quality of life of elderly people and challenging the efficiency and sustainability of healthcare systems. Non-communicable diseases (NCDs) are considered a global emergency responsible for over 70% of deaths worldwide. NCDs are also the basis for complex and multifactorial diseases such as hypertension, diabetes, and obesity. The epidemics of NCDs are a consequence of a complex interaction between health, economic growth, and development. This interaction includes the individual genome, the microbiome, the metabolome, the immune status, and environmental factors such as nutritional and chemical exposure. To counteract NCDs, it is therefore essential to develop an innovative, personalized, preventative, early care model through the integration of different molecular profiles of individuals to identify both the critical biomarkers of NCD susceptibility and to discover novel therapeutic targets.

## 1. Introduction

Precision medicine (PM) represents the most modern trend in contemporary medicine. On the basis of enormous amounts of data relating to people’s health, individual characteristics, and life context, PM aims to explore the human being as unique and unrepeatable, the individual mechanisms that determine the diseases, and the most appropriate strategies to prevent and cure them [1].

Recent developments in genetic sequencing techniques have allowed precision medicine to become a model for disease prevention, diagnosis, and treatment based on the patient’s genotype in many human genetic diseases. This model is applicable to non-communicable diseases (NCDs) and has been particularly implemented for the treatment of cancer [2].

NCDs are considered the most feared killer, responsible for 70% of all deaths, as well as the main threat to the sustainability of health systems. The main NCDs are cardiovascular disease, diabetes, cancer, neurodegenerative disorders, and chronic respiratory diseases. The World Health Organization compiled a list of the biggest threats to global health in 2019 and NCDs are considered the third-highest threat in order of gravity (Figure 1). In fact, NCDs include hypertension, diabetes, and obesity that affect millions of people around the world [3,4,5,6]. NCDs are caused by complex interactions between predisposing genetic traits (at the individual and population level) with other factors such as indiscriminate urbanization, pollution, climate change, unhealthy lifestyles, and aging (Figure 2) [7,8]. Therefore, a synergistic and structured approach to assess an individual risk, which includes both the genetic make-up and the evaluation of specific environmental factors (such as belonging to a particular socio-economic group), could be helpful in combating these diseases. It is estimated (source: WHO) that the implementation of policies to prevent and combat NCDs could translate into economic growth for low- and middle-income countries, up to USD 350 billion by 2030 [9]. The economic weight of NCDs must lead governments and institutions to take urgent measures to support research and enhancement of skills, the development of technological innovation, and the thrust of good social innovation practices. Governments should spend more on health education and in providing citizens with awareness and essential infrastructure and tools to make informed lifestyle and medical decisions towards maintaining good health. In addition, it is important to both improve care quality and encourage innovation to have a positive impact on health outcomes, and last but certainly not least, invest in prevention, and in particular, in preventive health care.

## 2. Shortfalls of Modern Medicine 

Over a quarter of drugs that enter clinical development fail because they are ineffective [10]. Other medicines that enter the health market are effective only in a fraction of patients. The consequence is the basis of failure in many cases of modern medical therapy. Many reasons are identified as causing disappointing results when using drugs outside clinical trials in actual clinical care. In early clinical development, more research and innovation are required to secure drug delivery and adequate drug dose exposure at the site of action; possibly, nanotechnology, modeling, and simulation within in silico trials may help in addressing the complexity of real drug exposure in a clinical setting.

In addition, in the past, most drugs developed in clinical trials were focused mostly on the average results in large populations that were “clinically” homogeneous. This approach is now profoundly challenged by the evolving taxonomy of diseases addressing disease heterogeneity (especially in cancer), along with clinical and genomic patient variability. It is therefore imperative to leverage new scientific advances and technologies, especially in the field of genetics, omics and immunology, to redefine the enrolment criteria and the design and conduct innovative and adaptive clinical trials. The outcomes measured in development are then to be repealed as a continuum in actual clinical care to provide the real-world data and much-needed evidence to define not only efficacy but effectiveness and actual value. The failure of many medical products also involves an economic problem as very expensive drugs have limited effectiveness [11]. It is estimated that genetic factors can account for 20 to 95 percent of the variability in drug disposition, efficacy, and adverse effects [12]. Looking at patients as a whole, nongenetic factors—including organ function, concomitant illnesses and therapies, drug interactions, and the complexity of chronic diseases—concur to limit the effectiveness of the drugs and to cause adverse reactions to traditional pharmacological therapies. The response to a drug is multifactorial and therefore variability is due to the described set of genetic and environmental factors interacting with each other. In this context, genetic factors can play an important role at various physiological levels: absorption, transport, metabolism, and excretion of molecules (pharmacokinetics), as well as in the interaction with the target and the rapport between the concentration and effect (pharmacodynamics) [13,14]. Genetic differences in these processes in genes coding enzymes, receptors, and proteins involved in these steps may have an effect on efficacy and toxicity [12].

Thanks to the knowledge acquired on the human genome after the year 2000, the results of studies on its variability, and the more efficient and affordable new sequencing functional technologies, it is possible to identify the sequences of genes involved in the activity and toxicity of drugs, and in some cases, define the genetic component driving individual response to drug treatment [15,16]. The analysis of inter-individual differences in DNA sequence and expression represents today an elective approach to understand and anticipate the variability of drug response. This is why the pharmaceutical industry, academia, clinicians, and regulators have focused their attention on the genomic basis of individual response variability. This has led to a transition from population-based to more individualized treatment both in clinical drug development and clinical practice [17]. This concept is the basis of pharmacogenomics, a crucial pillar of “personalized medicine” that studies genomic and epigenomic factors that influence drug pharmacokinetics (PK) [18] and pharmacodynamics (PD), as well as drug–drug interactions (DDIs). Some examples are cytochrome P450 enzymes and CYPs (drug PK), cancer, HIV, and cystic fibrosis (drug efficacy), and QT prolongation (drug safety) [19,20,21,22]. Knowledge of genomic variation is now translated into clinical application by the introduction of pharmacogenomic information into the summaries of product characteristics (SmPCs). Explicit recommendations on when and how to include pharmacogenomic information in the SmPC are provided in a number of regulatory guidelines: the European Commission SmPC guideline (rev. 2009) [23], the EMA guideline on the use of pharmacogenetic methodologies in the pharmacokinetic evaluation of medicinal products (2009) [18], and the guideline on key aspects for the use of pharmacogenomics in the pharmacovigilance of medicinal products (2013) [24]. Prescribers are informed of whether predictive genotyping of the patient’s germline or somatic DNA is mandatory, highly recommended, or informative. Up to 2015, the SmPCs of 150 medicines (about 30% of all new medicinal products) approved by the European Medicines Agency included pharmacogenomic information, and for the US Food and Drug Administration, similar numbers apply [25]. As further discussed below, genome-driven biomarkers have been proven actionable and clinically useful in a variety of clinical conditions to allow a more patient-centered and precise medicine prescription [10,26]. Genomic biomarkers are currently required for both treatment selection and/or exclusion; particularly, cancer status (e.g., evidence of wild-type RAS (KRAS and NRAS) is required before initiating treatment with Erbitux in metastatic colon cancer expressing epidermal growth factor receptor (EGFR); in the treatment of epilepsy to prevent patient’s hypersensitivity reactions (HLA-B*1502 in Han Chinese and Thai patients strongly predicts the risk of severe carbamazepine-associated Stevens–Johnson syndrome) [27]; and for adapted dose determination (e.g., use of Mercaptopurine in acute lymphoblastic leukemia: patients with homozygous deficiency of either TPMT OR NUDT15 enzyme require 10% or less of the standard oral suspension dose) [28].

## 3. Genomic Biomarkers in NCDs

The expanding knowledge of the genetic basis, pathogenesis, and therapeutic possibilities of NCDs has provided new targets for early diagnosis of human diseases, drug discovery, and drug development and suggested strategies to translate this knowledge into clinical practice as (initial) clinical trials. Biomarkers represent a measurable biological indicator able to provide information on risk assessment, diagnosis, and therapy for NCDs. Omic sciences (genomic, transcriptomics, proteomics, metabolomics) offer opportunities to identify novel biomarkers for defining and understanding the molecular basis of NCDs. A genomic biomarker (GB) is defined as a measurable DNA or RNA characteristic that is an indicator of normal biologic processes, pathogenic processes, and/or response to therapeutic or other intervention [29]. A genomic biomarker could, for example, reflect the regulation of a gene, the expression of a gene, or the function of a gene. A genomic biomarker does not include proteomics and/or metabolomics. GBs in the last 25 years have revolutionized the study of NCDs, defining individual risks, population risks, pathogenesis, prognosis, and, recently, therapy. Many of these GBs have been identified by GWAS studies [30]. GBs have also allowed us to better understand the links between genotype and environment, opening a new line of research, epigenetics [28]. Epigenetics and high-throughput omics sequencing have indeed started unveiling mechanisms connecting multi-organ response, among others, to exercise, nutrition, relational and emotional, environmental stressors [31]. These intersecting sciences are producing evidence highly complex and there are often missing points of connection that require yet more research efforts. Most importantly, the scientific community is now embracing the challenge of starting the development of reference standards in order to ensure the robustness and reproducibility of the results in different studies [32,33]. It is now generally accepted that the most promising and sustainable approach for the development of disease treatments and prevention should consider individual variability in genes, environment, and lifestyle for each individual, as stated by NIH [32,33]. This, in turn, requires the definition of patient groups/individualization to decide on whether to treat or what dose to use to estimate response and toxicity to a given treatment, decide what “drug-cocktail” is most “promising”, and finally which companion diagnostics (CDx) is needed for GB detection. Patients with particular *EGFR* mutations have better survival status than those without, indicating the benefit of TKI therapy. Individuals harboring somatic mutations in *TP53*, *LRP1B*, *STK11*, *KEAP1*, *BRAF*, *MET*, and *MRC2* had significantly shorter survival time than individuals with wild-type, which suggest that mutations in these genes can be used as prognostic GBs in clinical practice [34]. In oesophageal cancer, molecular signatures have allowed researchers to identify three distinct molecular subtypes with potential therapeutic relevance [35]. In psoriasis, *HLA-C*06:02* genotype is a predictive GB of biologic treatment response. *HLA-C*06:02*–negative patients respond better to adalimumab than ustekinumab [36,37]. Other examples of NCDs which involve the use of GB for diagnosis or choice of therapy include cardiovascular disorders, diabetes, Parkinson’s disease, age-related macular degeneration, inflammatory bowel diseases, autism spectrum disorder, and schizophrenia (Table 1) [38,39,40,41,42,43,44]. These examples are set to increase in the coming years according to the discovery of rare alleles [45]. For example, very recently, Dwivedi et al. (2019) demonstrated that a rare loss-of-function allele, p.Arg138*, in the *SLC30A8* gene encoding the zinc transporter 8 (ZnT8), which is enriched in Western Finland, protects against type 2 diabetes (T2D) [46]. This last example represents a model in our opinion of the translation of a GB discovered during research to clinical practice. A good model for which GBs affect diagnosis and therapy is represented by rare diseases and, in particular, neuromuscular diseases (NMDs) [47]. In fact, clinical trials of new therapies are made more difficult due to lack of reliable and monitorable clinical outcome measures. Genomic biomarkers have and will have a way to speed up research in this field, shedding light on the pathophysiological mechanisms behind such diseases and providing invaluable tools for monitoring their progression, prognosis, and response to drug treatment. Furthermore, genomic biomarkers represent a surrogate endpoint for clinical trials, enabling better stratification of patient cohorts through more accurate diagnosis and prognosis prediction.

The research in NCD precision medicine has intensified in the last decade, revealing many new and important GBs that determine risk assessment, drug responses, and prognosis. Genetic variants to be tested are often identified according to their occurrence in defined populations (i.e., Caucasians, Hispanics, Asians, Arabs, Africans). This could lead to a Bayesian error, given the fact that a considerable proportion of the genetics of the population is influenced by migration, and therefore, a selection of GBs according to population prevalence could be inappropriate in the European context. One well designed and conclusive randomized pharmacogenomic trial was the PREDICT-1 study encompassing 1956 patients from 19 countries, revealing the very specific influence of HLA-B*5701 on the toxicity of abacavir [48].

A significant proportion of the clinical studies involving genetics and genomics have been inconclusive or non-replicable for various methodological flaws which include

Poor quality of test samples;Analyzing somatic instead of germline DNA;Analyses of non-relevant GBs;Poor quality of the employed analytics;Lack of appropriate phenotype identification;Inadequate study design, including lack of appropriate patient selection/stratification;Lack of statistical analysis planning and execution in relation to the frequency of GB in the population.

These problems very often arise from the lack of standardization of novel genomic biomarkers and, therefore, from the non-observance of the validation and qualification processes. Genomic biomarkers require methodological validation including being (a) repeatable within a laboratory, (b) reproducible across laboratories, (c) concordant across platforms, (d) comparable with alternative confirmatory technologies, and (e) reflective of biology regardless of the differences in technology. Genomic biomarkers also require a qualification step [49]. This is assured by regulatory agencies (e.g., FDA and EMA) [50]. It is not simple for researchers to follow regulatory guidelines and rules produced by regulatory agencies to qualify an identified biomarker. Therefore, it is essential to have specialists who can guide, coordinate, and address researchers in this kind of work and to keep the contacts between them and regulatory agencies.

## 4. Conclusions and Future Challenges

The epidemic of NCDs is the consequence of the complex interactions between individual and population genetic factors and universal trends such as the aging of the population, indiscriminate urbanization, climate change, and the worldwide spread of unhealthy lifestyles [6,7]. All this requires a radical change in the models of care used so far in the world and the development of new prevention and intervention protocols at a global level with new technologies, new educational systems, new financial instruments, and new forms of collaboration and interactions between the various actors (governments, universities, patient associations, industry, civil society). In most countries, national health spending is expected to continue to grow faster than GDP; at current levels and at the rate of growth, healthcare costs will make many systems unsustainable in the near future. In developing countries, these costs will be unsustainable even for the availability of a drug such as insulin, which has been known for many years. The cost of insulin has tripled in recent years instead of decreasing. Recent innovations have brought significant improvements to health outcomes, but have also led to new tools and services that increase healthcare costs. Furthermore, as the NCD population accounts for the majority of health expenditure, better standardization of health services along with more effective and more integrated patient management and prevention strategies could substantially reduce costs and improve quality. There is no doubt that prevention, both primary and secondary prevention, is the only long-term answer to the problem of sustainability. Thus, innovative health strategies for the management of NCDs is of the utmost importance to prevent the impact of NCDs on public health.

Future multi-omics studies will contribute to addressing the clinical challenges, particularly the identification of molecular signatures and subtype-specific phenotypes able to distinguish between the currently broadly defined conditions, with more precise levels of personalization and prediction of health outcomes. Future research efforts will significantly contribute to identifying clinically actionable genomic biomarkers contributing to critical improvements in prevention, early diagnosis, the likelihood of response to intervention, “precision” medicines, and prognosis.

## Figures and Tables

**Figure 1 high-throughput-09-00003-f001:**
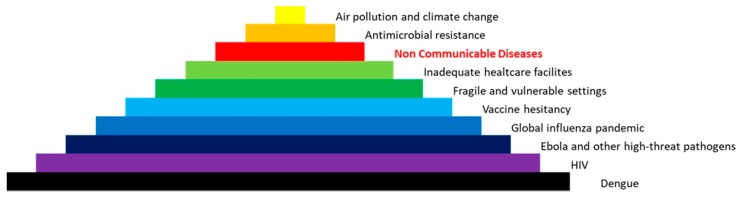
Top ten threats to global health in 2019. List of the biggest threats to global health in 2019 compiled by the World Health Organisation and ranked by a community of healthcare professionals. Non-communicable diseases (NCDs) are listed as a serious threat.

**Figure 2 high-throughput-09-00003-f002:**
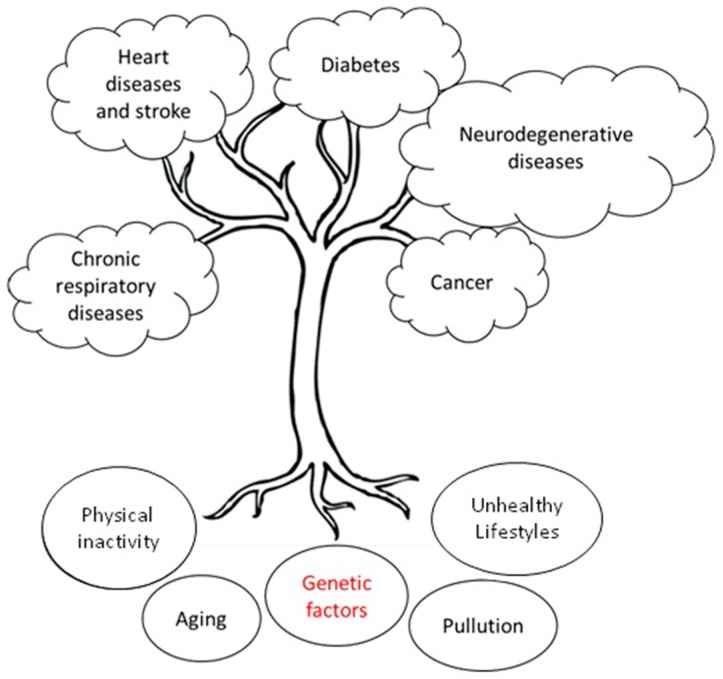
Main NCDs and principal factors (roots) most contributing to their development. NCDs are caused by complex interactions between predisposing genetic traits with environmental factors.

**Table 1 high-throughput-09-00003-t001:** Examples of genes involved in response to treatment of some NCDs.

Diseases	Drugs	Genes
Psoriatic Arthritis	Methotrexate	*DHFR, MTHFR*
	Sulfasalazine	*NAT2*
	Etanercept	*TNF-α, FCGR2A*
	Infliximab	*TNFRSF1A*
	Adalimumab	*HLA*
Parkinson’s Disease	Levodopa	*COMT, DDC, DRD2, DRD3 SLC6A3, SLC22A1, APOE, ACE, CCK, BDNF, HCRT, OPRM1, HOMER1, SV2C, GRIN2A, ADORA2A*
	Tolcapone/entacapone	*UGT1A6, COMT*
	Ropinirole/pramipexole	*CDRD2, DRD3*
	Rasagiline	*DRD2*
Age-Related Macular Degeneration	Ranibizumab	*VEGFA, CFH, ARMS, VGFR2, APOE, CD36*
	Bevacizumab	*VEGFA, CFH, ARMS*
	Aflibercept	*CFH, ARMS*
	Verteporfin	*CRP*
Crohn’s disease	Aminosalicylates	*HLA*
	Immunosuppressors	*TPMT, ITPA, NUDT15, ABCC4, GST, HLA, ABBC4, ABCB1, RAC1, MTHFR*
	Glucocorticoids	*NR3C1, FKBP5, ABCB1, TNF, NLRP1*
	Infliximab	*ADAM17, IL1β, TNF-α, TNFRSF1A, TNFRSF1B, FASL, CASP9, FCGR3A, ATG16L1*
	Adalimumab	*ATG16L1, HFE*
Autism Spectrum Disorder	Risperidone	*CYP2D6, ABCB1, DRD2, DRD3, HTR2A, HTR2C, FTO, MC4R, LEP, CNR1, FAAH*
	Fluvoxamine	*SLC6A4*
	Escitalopram	*SLC6A4, CYP2C19*
	Methylphenidate	*DRD1–DRD5, ADRA2A, SLC6A3, SLC6A4, MAOA, MAOB, COMT*
Schizophrenia	Clozapine	*HTR2A, HTR3A, DRD2, DRD4, COMT, CYP2C19, SLC6A9, ITIH3*
	Risperidone	*ABCB1, GRID2, GRM7, GRM5*
	Amisulpride	*SNAP25, ANKS1B*
	Paliperidone	*ADCK1*
	Pomaglumetad methionil	*HTR2A*

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
