# Peer review of "Precision Medicine in Non-Communicable Diseases"

_2571-5135, 2020, doi:10.3390/ht9010003_

Round 1

Reviewer 1 Report

There are several typographic and grammatical errors in the text and the figures that need to be corrected

References should be added to the part referring to the percentage of NCDs globally at line 42. 

The authors should define what kind of measures should the governments take to reduce the economic burden of NCDs that is mentioned in lines 54-56

The authors should explain the reasons why traditional drugs are inefficient in the treatment of many NCDs as it is mentioned in line 60

References should be added on the roles of genetic factors in pharmacokinetics and pharmacodynamics in lines 72-74

Most importantly precision medicine is based on several omic sciences including proteomics and metabolomics especially in the case of NCDs and the paper is focused exclusively on the role of genomics in Precision Medicine. The authors are only mentioning the existence of the other fields but do not write any details about them. It is well established in the last years that environment and lifestyle have a major effect on proteomics and metabolomics, therefore, they need to be an important part of precision medicine (lines 139-140). The authors should write a part focused on these.

Is there a pattern of genomic biomarkers for important diseases based on research studies? Is there any consistency between studies that could allow the establishment of certain genomic biomarkers?

The authors are mentioning the methodological flaws of clinical studies that result in inconsistencies and inconclusive results in lines 182-192. Which are the reasons for these flaws and what are the steps that need to be taken in the future to solve these problems and eliminate the flaws?

Author Response

To Reviewer 1: 

Q1): There are several typographic and grammatical errors in the text and the figures that need to be corrected

Answer: Thanks, typographical errors in the text and figures have been corrected.  

Q2): References should be added to the part referring to the percentage of NCDs globally at line 42. 

Answer: References have been revised and suitably integrated and updated.

Q3): The authors should define what kind of measures should the governments take to reduce the economic burden of NCDs that is mentioned in lines 54-56:

Answer: Thanks for this tip. We have appropriately inserted the following in the text:

Governments should spend more in  health education and in providing citizens with the awareness and the essential infrastructures and tools to make informed lifestyle and medical decisions towards maintaining good health. In addition, it is important to both improve care quality and encourage innovation to have a positive impact on health outcomes. Last but certainly not least, invest in prevention, and in particular in the primary one.

Q4): The authors should explain the reasons why traditional drugs are inefficient in the treatment of many NCDs as it is mentioned in line 60

Answer: Thanks for this tip. We have appropriately inserted the following in the text:

Many reasons are identified as causing disappointing results when using drugs outside clinical trials in to actual clinical care.  In early clinical development more research and innovation is required to secure  drug delivery and adequate drug dose exposure at site of action: possibly nanotechnology, modeling and simulation  within in silico trials may help in addressing the complexity of real drug exposure in clinical setting. In addition in the past most drugs have been developed in clinical trials that were focusing mostly on the average results in large populations homogeneous “clinically”. This approach is now profoundly challenged by the evolving taxonomy of diseases addressing disease heterogeneity (re especially in cancer),along with clinical and genomic patient variability. It is therefore imperative to leverage new scientific advances  and technologies especially in the field of genetics, omics and immunology, to redefine the enrolment criteria and the design and conduct innovative and adaptive clinical trials. The outcomes measured in development actually are then to be repealed as a continuum in actual clinical care to provide the real world data and evidence much needed to define not only efficacy but effectiveness and actual value.

Q5): References should be added on the roles of genetic factors in pharmacokinetics and pharmacodynamics in lines 72-74

Answer: The appropriate references have been inserted and the context sentences modified

Q6): Most importantly precision medicine is based on several omic sciences including proteomics and metabolomics especially in the case of NCDs and the paper is focused exclusively on the role of genomics in Precision Medicine. The authors are only mentioning the existence of the other fields but do not write any details about them. It is well established in the last years that environment and lifestyle have a major effect on proteomics and metabolomics, therefore, they need to be an important part of precision medicine (lines 139-140). The authors should write a part focused on these.

 Answer: Thanks for this important tip. The context sentence has been substantially modified as reported below and appropriate references have been inserted.

…Epigenetics and high-throughput omics sequencing indeed start unveiling mechanisms connecting multi-organ response, among others, to exercise, nutrition, relational and emotional environmental stressors.  These intersecting sciences, are producing evidence highly complex and often there are missing points of connection that require yet more research efforts. Most importantly the scientific community is now embracing the challenge of starting the development of reference standards in order to ensure robustness and reproducibility of the results in different studies [33].

Q7): Is there a pattern of genomic biomarkers for important diseases based on research studies? Is there any consistency between studies that could allow the establishment of certain genomic biomarkers?

Answer: Thanks for this important tip. The context sentence has been substantially modified as reported below and appropriate references have been inserted.

This last example represents a model in our opinion of translation of a GB discovered during a research to clinical practice. A good model for which GBs affected diagnosis and therapy is represented by rare diseases and in particular by neuromuscular diseases (NMDs) [47]. In fact, clinical trials of new therapies are made more difficult due to lack of reliable and monitorable clinical outcome measures. Genomic biomarkers have and will have a way to speed up research in this field, shedding light on the pathophysiological mechanisms behind such diseases and providing invaluable tools for monitoring their progression, prognosis and response to drug treatment. Furthermore, genomic biomarkers represent a surrogate endpoint for clinical trials, enabling better stratification of patient cohorts through more accurate diagnosis and prognosis prediction.

Q8): The authors are mentioning the methodological flaws of clinical studies that result in inconsistencies and inconclusive results in lines 182-192. Which are the reasons for these flaws and what are the steps that need to be taken in the future to solve these problems and eliminate the flaws?

Answer: Thanks for this important tip. The context sentence has been substantially modified as reported below and appropriate references have been inserted.

These problems very often arise from the lack of standardization of novel genomic biomarkers and therefore from the non-observance of the validation and qualification processes. Genomic biomarkers require a methodological validation including: a) repeatable within a laboratory; b) reproducible across laboratories; c) concordant across platforms; d) comparable with alternative confirmatory technologies, e) reflective of biology regardless of the differences in technology. Genomic biomarkers require also a qualification step [49]. This is assured by regulatory agencies (e.g. FDA and EMA) [50]. It is not simple for researchers to follow regulatory Guidelines and rules produced by regulatory agencies to qualify an identified biomarker. Therefore, it is essential to have specialists who can guide, coordinate and address researchers in this kind of work and to keep the contacts between them and regulatory agencies.

Reviewer 2 Report

The topic covered by the review is very interesting.
However, the matter should be more examined in depth, for example with tables and / or graphs referred to the different diseases, belonging to the NCDs group, where precision medicine was applied.
Furthermore, the review makes specific reference to genomics, what is known for other OMIC sciences?

Author Response

To Reviewer 2

Q): However, the matter should be more examined in depth, for example with tables and / or graphs referred to the different diseases, belonging to the NCDs group, where precision medicine was applied.

Answer: Thanks for this suggestion. A table reporting examples of genes involved in response to treatment of some NCDs has been now included in the paper.

Q2): Furthermore, the review makes specific reference to genomics, what is known for other OMIC sciences?

In our paper we have focused only on genomic biomarkers which  according to the ICH (International Conference of Harmonization) does not involve biomarkers derived from proteomics or metabolomics. However, in this new version of the manuscript, we have added some ad hoc sentences and integrated the bibliography (see items 47-50).

Round 2

Reviewer 2 Report

The manuscript has been sufficiently modified according to the raised comments